# Multi-Axis Laser Interferometer Not Affected by Installation Errors Based on Nonlinear Computation

**Tingrui Wen** **, Jinchun Hu \*, Yu Zhu and Guojie Hua**

Department of Mechanical Engineering, Tsinghua University, Beijing 100084, China; wentr20@mails.tsinghua.edu.cn (T.W.); zhuyu@mail.tsinghua.edu.cn (Y.Z.); hgj19@mails.tsinghua.edu.cn (G.H.)
\* Correspondence: hujinchun@tsinghua.edu.cn

**Abstract:** Most multi-axis laser interferometers require accurate installation of the lasers and detectors since the position and orientation installation errors of lasers and detectors bring measurement error to the interferometer. In this paper, the multi-axis interferometer based on nonlinear computation is proposed, which avoids the measurement error caused by installation errors by taking the laser position and orientation as unknowns into the measurement model and discussing the solution component uniqueness of a nonlinear equation system. The simulation results show that even when the installation errors are close to 1 μm and 1 μrad, the proposed interferometer can still measure the multi-degree-of-freedom displacement accurately, and the root mean standard error (RMSE) of displacement is 1.884 nm and $5.871 \times 10^{-7}$ mrad under a reading noise level of 0.1 nm.

**Keywords:** multi-axis laser interferometer; installation errors; solution component uniqueness

## 1. Introduction

With a displacement measurement resolution higher than 1 nm, the laser interferometer is used in various precision and ultra-precision measurement applications. One of the typical applications is the lithography wafer stage in semiconductor equipment, which is used to carry the silicon wafer to achieve a series of multi-degree-of-freedom (multi-DOF) precision motions. Since the positioning accuracy of the wafer stage directly affects the overlay accuracy of the lithography, a high requirement for the accuracy and bandwidth of the multi-DOF displacement measurement is put forward [1–3].

The laser interferometer is commonly used in such cases to measure displacement based on the interference of laser beams, which measure the variation in optical path length by detecting the laser intensity, thus realizing the precise displacement measurement [4–6]. According to the laser type, the interferometers can be classified as homodyne interferometers with a single-frequency laser source and heterodyne interferometers with a dual-frequency laser source. To measure the multi-DOF displacement of a precision movement stage by using a multi-axis interferometer, more lasers and other optical elements need to be adopted. Additionally, it is necessary to carefully design the number and each installation position of lasers and the multi-DOF displacement decoupling computation. For example, five lasers are used to measure the stage surface at five different points in the five-axis laser interferometer, and the 5-DOF displacement is computed from the interferometer model [7]. In addition to the measurement scheme in [7], some researchers used more lasers with redundant information to decrease the nonlinear error [8–11]. Moreover, some researchers built a more accurate but more complex model of the coupling between multi-DOF displacement to improve the accuracy of the displacement computation results [12–15].

However, these multi-axis laser interferometers mentioned above inevitably have measurement systematical errors, which can be summarized as two reasons. First, the interferometer model is inaccurate, which is mainly caused by installation errors. For example,

the laser beams in the model are parallel, but in practice they are not, and the installation position and orientation of the lasers in the model are different from those in practice. Second, the computation of the model is inaccurate. Usually, there are many nonlinear functions such as trigonometric functions and rational fractions in the interferometer model. To simplify the computation, these nonlinear functions are approximated linearly so that the explicit expression of the multi-DOF displacement can be derived. Both of the above two reasons will result in a multi-DOF displacement measurement error, which causes the loss of accuracy.

To avoid the above two factors leading to a measurement systematical error, parameters such as the laser beam angle and emitting point position should be introduced into the model as unknowns. In addition, the nonlinear measurement model should be solved directly without any linear approximation. However, in this way, the interferometers model is quite complex; that is, it contains a large number of unknowns and complex nonlinear functions, and how to solve the model accurately and quickly becomes a new problem. Although there are the Gauss–Newton method [16], Levenberg–Marquardt (LM) method, and other nonlinear equation system numerical algorithms [17–19], when it comes to practical engineering, due to the negative properties of the equations such as the singular Jacobi matrix, which are caused by the huge number of unknowns and the complex nonlinear functions, the convergence and the accuracy of the algorithm may not be ideal, which raises the necessity for a general and comprehensive analysis.

In this paper, an interferometer model with unknown parameters, including the laser beam angle and emitting point position, is proposed to avoid inaccurate modeling so that the measurement result is entirely unaffected by installation errors. A solution of the interferometer model without any linearization is proposed to avoid an inaccurate solution. For the problem arising in the model solving that the singular Jacobi matrix leads to no unique solution of the equations, the theory of solution component uniqueness is proposed to obtain a unique and accurate solution of the displacement in the model. This paper is organized as follows. In Section 2, the mathematical model of the laser interferometer is proposed, which shows the relation between interferometer readings and multi-DOF displacements. In Section 3, the displacement computation principle is proposed, and the focus problem that the solution of a nonlinear equation system is not unique due to the singular Jacobi matrix is discussed. In Section 4, the interferometer measurement and the displacement computation are simulated. The simulation result verifies the effectiveness of the proposed interferometer. Section 5 is the conclusion of the paper.

## 2. Interferometer Measurement Model

In this section, the structure and the mathematical model of the multi-axis laser interferometer, which avoids the two factors mentioned in Section 1, is proposed. For simplicity, the 3-DOF displacement measurement of the X-Y stage is mainly discussed, while the 6-DOF displacement measurement of the X-Y-Z stage is also mentioned, but is not the focus. Nevertheless, there is no difference in the principle between the two but there is in the computation complexity.

### 2.1. Single-Axis Interferometer Laser Path Model

Since the multi-axis interferometer consists of multiple regular single-axis interferometers, the single interferometer model is proposed first.

The measurement principle of the single interferometer is shown in Figure 1; the laser emits from the interferometer, reflects off the smooth surface of the stage (or the retroreflector fixed on the stage), and returns to the interferometer. Let a fixed position of the stage be the reference position, as shown in Figure 1a, and the reading of the interferometer at the reference position is set to 0. Then, the reading of the interferometer at the measurement position in Figure 1b is set as *V* and is equal to the variation of the

laser path length after the stage is moved from the reference position to the measurement position, that is

$$V = L_1 - L_0 \tag{1}$$

where $L_0, L_1$ are, respectively, the laser path length at the reference and measurement position.

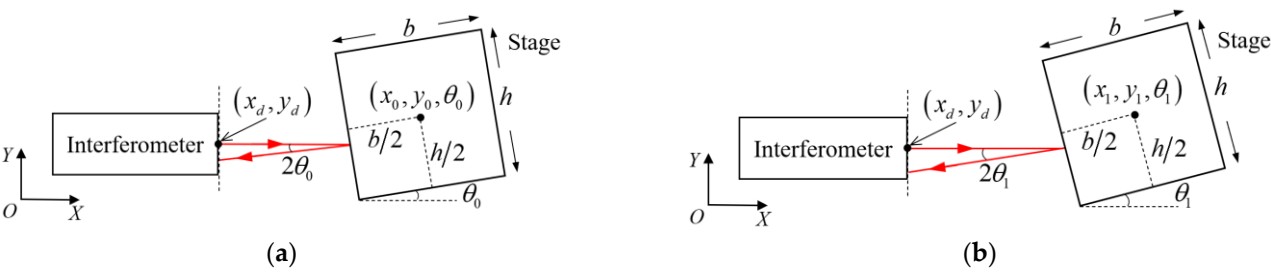

**(a)**                                   **(b)**

**Figure 1.** (**a**) Interferometer and stage at reference position. (**b**) Interferometer and stage at measurement position.

Moreover, it can be obtained from Figure 1 that

$$\begin{cases} L_0 = \frac{2\cos\theta_0}{\cos 2\theta_0}\left[(x_0 - x_d)\cos\theta_0 + (y_0 - y_d)\sin\theta_0 - b/2\right] \\ L_1 = \frac{2\cos\theta_1}{\cos 2\theta_1}\left[(x_1 - x_d)\cos\theta_1 + (y_1 - y_d)\sin\theta_1 - b/2\right] \end{cases} \tag{2}$$

where $(x_d, y_d)$ is the coordinate of the laser emission point, $(x_0, y_0)$ and $(x_1, y_1)$ are, respectively, the coordinate of the stage center at the reference and measurement position, and $\theta_0, \theta_1$ are, respectively, the angle of the stage at the reference and measurement position. $b$ and $h$ are the length and width of the stage.

The single interferometer model proposed in Equations (1) and (2) shows the mathematical relation between the interferometer readings and the position of the stage. This model is still used in the following multi-axis interferometers.

*2.2. 3-DOF Displacement Measurement Model*

To avoid measurement errors caused by installation errors, the installation parameters such as laser beam angles and emission point positions are added to the model as unknowns. Then, it is necessary to add constraints to make the model solvable, and one way is to increase the redundant measurement information by using more interferometers.

In this paper, a 3-DOF displacement measurement of the X-Y stage using a six-axis interferometer is proposed. And the measurement scheme is shown in Figure 2. There are six interferometers for measuring the two adjacent sides of the X-Y stage. Assuming that the X-Y stage only moves in the X-O-Y plane, and all the laser beams are supposed to be in the X-O-Y plane, the measurement scheme is also expressed in Figure 3.

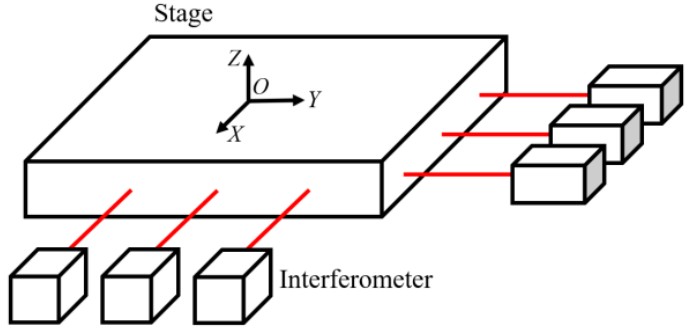

**Figure 2.** Scheme of 3-DOF displacement measurement.

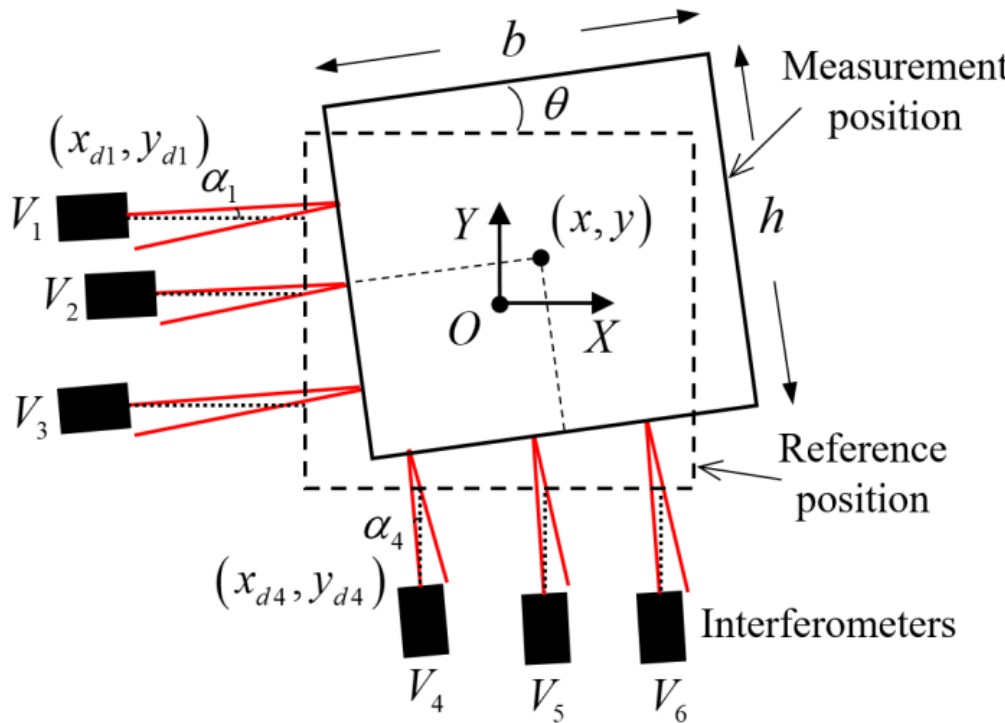

**Figure 3.** Measurement principle of 3-DOF displacement of X-Y stage with a 6-axis interferometer.

In Figure 3, the reference position of the stage is shown as the dotted line, and the measurement position is shown as the true line; the O-XYZ axis is set so that the coordinates of the stage center and the angle of the stage are both equal to 0 at the reference position. Due to the installation errors, there is an angle of each laser beam, which is set as $\alpha_i, i = 1, 2, \ldots, 6$, and the unknown installation position of each interferometer is set as $(x_{di}, y_{di}), i = 1, 2, \ldots, 6$.

Based on the single interferometer laser path model proposed in Section 2.1, the readings of the six interferometers can be modeled as

$$
\begin{cases}
V_1 = \frac{2\cos(\theta-\alpha_1)}{\cos(2\theta-2\alpha_1)}[(x-x_{d1})\cos\theta + (y-y_{d1})\sin\theta - b/2] + \frac{2\cos\alpha_1}{\cos 2\alpha_1}(x_{d1}+b/2) \\
V_2 = \frac{2\cos(\theta-\alpha_2)}{\cos(2\theta-2\alpha_2)}[(x-x_{d2})\cos\theta + (y-y_{d2})\sin\theta - b/2] + \frac{2\cos\alpha_2}{\cos 2\alpha_2}(x_{d2}+b/2) \\
V_3 = \frac{2\cos(\theta-\alpha_3)}{\cos(2\theta-2\alpha_3)}[(x-x_{d3})\cos\theta + (y-y_{d3})\sin\theta - b/2] + \frac{2\cos\alpha_3}{\cos 2\alpha_3}(x_{d3}+b/2) \\
V_4 = \frac{2\cos(\theta-\alpha_4)}{\cos(2\theta-2\alpha_4)}[(y-y_{d4})\cos\theta + (x-x_{d4})\sin\theta - h/2] + \frac{2\cos\alpha_4}{\cos 2\alpha_4}(y_{d4}+h/2) \\
V_5 = \frac{2\cos(\theta-\alpha_5)}{\cos(2\theta-2\alpha_5)}[(y-y_{d5})\cos\theta + (x-x_{d5})\sin\theta - h/2] + \frac{2\cos\alpha_5}{\cos 2\alpha_5}(y_{d5}+h/2) \\
V_6 = \frac{2\cos(\theta-\alpha_6)}{\cos(2\theta-2\alpha_6)}[(y-y_{d6})\cos\theta + (x-x_{d6})\sin\theta - h/2] + \frac{2\cos\alpha_6}{\cos 2\alpha_6}(y_{d6}+h/2)
\end{cases}
\tag{3}
$$

where $V_i$ is the reading of the $i$th interferometer, $i = 1, 2, \ldots, 6$, and $(x, y, \theta)$ is the 3-DOF displacement of the stage at the measurement position since the coordinate of the stage center is equal to 0 at the reference position.

According to Equation (3), after obtaining the value of $V_i$ in practice, there are a total of 21 unknowns including $(x_{di}, y_{di}, \alpha_i), i = 1, 2, \ldots, 6$, and $(x, y, \theta)$ in the nonlinear equation system. Since the number of unknowns is greater than the number of constraints, it is unrealistic to solve the displacement $(x, y, \theta)$ from Equation (3). Therefore, the number of measurements is increased to gain more redundant information; that is, all the interferometers' readings are recorded at every 10 distinct measurement positions for one displacement computation. Let $(x_n, y_n, \theta_n)$ be the displacement at $n$th measurement posi-



tions, $n = 1, 2, \ldots, 10$, the six interferometers' readings at 10 measurement positions can be modeled as

$$
\begin{cases}
V_{n1} = \frac{2\cos(\theta_n - \alpha_1)}{\cos(2\theta_n - 2\alpha_1)}[(x_n - x_{d1})\cos\theta_n + (y_n - y_{d1})\sin\theta_n - b/2] + \frac{2\cos\alpha_1}{\cos 2\alpha_1}(x_{d1} + b/2) \\
V_{n2} = \frac{2\cos(\theta_n - \alpha_2)}{\cos(2\theta_n - 2\alpha_2)}[(x_n - x_{d2})\cos\theta_n + (y_n - y_{d2})\sin\theta_n - b/2] + \frac{2\cos\alpha_2}{\cos 2\alpha_2}(x_{d2} + b/2) \\
V_{n3} = \frac{2\cos(\theta_n - \alpha_3)}{\cos(2\theta_n - 2\alpha_3)}[(x_n - x_{d3})\cos\theta_n + (y_n - y_{d3})\sin\theta_n - b/2] + \frac{2\cos\alpha_3}{\cos 2\alpha_3}(x_{d3} + b/2) \\
V_{n4} = \frac{2\cos(\theta_n - \alpha_4)}{\cos(2\theta_n - 2\alpha_4)}[(y_n - y_{d4})\cos\theta_n + (x_n - x_{d4})\sin\theta_n - h/2] + \frac{2\cos\alpha_4}{\cos 2\alpha_4}(y_{d4} + h/2) \\
V_{n5} = \frac{2\cos(\theta_n - \alpha_5)}{\cos(2\theta_n - 2\alpha_5)}[(y_n - y_{d5})\cos\theta_n + (x_n - x_{d5})\sin\theta_n - h/2] + \frac{2\cos\alpha_5}{\cos 2\alpha_5}(y_{d5} + h/2) \\
V_{n6} = \frac{2\cos(\theta_n - \alpha_6)}{\cos(2\theta_n - 2\alpha_6)}[(y_n - y_{d6})\cos\theta_n + (x_n - x_{d6})\sin\theta_n - h/2] + \frac{2\cos\alpha_6}{\cos 2\alpha_6}(y_{d6} + h/2)
\end{cases}
\tag{4}
$$

where $V_{ni}$ is the reading of the *i*th interferometer at the *n*th measurement position, $n = 1, 2, \ldots, 10$, $i = 1, 2, \ldots, 6$.

The equation system (4) is the proposed 3-DOF displacement measurement model. As we can see, the number of equations is 60, which is larger than the number of unknowns (48 in total, including $(x_n, y_n, \theta_n)$, $n = 1, 2, \ldots, 10$ and $(x_{di}, y_{di}, \alpha_i)$, $i = 1, 2, \ldots, 6$). The 3-DOF displacement can be computed from Equation (4) by using the nonlinear equations system algorithm, which is discussed in Section 3.

From Equation (4), it should be noted that since the installation positions and angles $(x_{di}, y_{di}, \alpha_i)$ are unknowns, the installation errors will not bring errors to the displacement computation. In addition, there is no linear approximation in Equation (4), which also makes the displacement computation result accurate.

It should also be noted that the number of interferometers and the number of measurement positions in the measurement model can be varied, and their selection rule is as follows: let the number of interferometers be $M$ and the number of measurement positions be $N$ (in Equation (4), $M = 6$ and $N = 10$). According to the above analysis, there are $MN$ equations and $(3M + 3N)$ unknowns in the model. To make the number of unknowns not greater than the number of equations, there is

$$
3M + 3N \leq MN
\tag{5}
$$

It can be seen from Equation (5) that $M$ is supposed to be larger than 3, otherwise the number of equations will never catch up with the number of unknowns. Hence, Equation (5) can be rewritten as

$$
N \geq 3 + \frac{9}{M - 3}
\tag{6}
$$

Equation (6) shows that $N$ can be smaller with larger $M$. Under the restriction of Equations (5) and (6), less $MN$ means less computational complexity, but large $MN$ provides enough redundant information to solve the unknowns. Therefore, it is selected that $M = 6$ and $N = 10$ in this section.

### 2.3. 6-DOF Displacement Measurement Model

In the 3-DOF displacement measurement of the X-Y stage, the assumption that the stage only moves in one plane is adopted. Although this assumption holds approximately in the case that the assembly is ideal, there is still deviation from the actual situation. When it comes to having sufficient computing resources, it is necessary to discuss and apply the 6-DOF displacement measurement to reduce the deviation between the model and the actual situation.

The 6-DOF displacement measurement scheme of the X-Y-Z stage using a nine-axis interferometer is proposed and shown in Figure 4. There are three interferometers that measure each one of the three adjacent surfaces of the stage. Similar to the 3-DOF displacement measurement of the X-Y stage, the mathematical model of the readings of the nine interferometers can be deduced, which is too complex because it needs to calculate the laser

path in 3-dimensional space. Since the 6-DOF model is only more complex than the 3-DOF model, and there is no difference in the modeling principle, the detailed expression of the 6-DOF model is not given in this section, but the function heading is given.

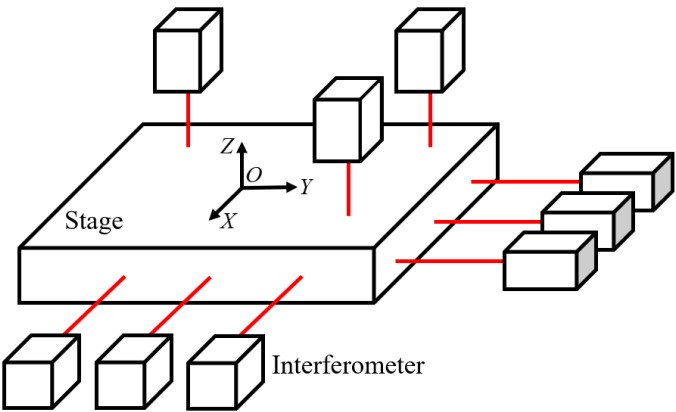

**Figure 4.** Scheme of 6-DOF displacement measurement.

The same as in the 3-DOF measurement, the O-XYZ coordinate axis is set to ensure that the origin coincides with the center of the stage at the reference position. Let $(x, y, z, \theta, \varphi, \omega)$ be the 6-DOF displacement, then the readings of interferometers at the measurement position are

$$V_i = V_i(x, y, z, \theta, \varphi, \omega, x_{di}, y_{di}, z_{di}, \alpha_i, \beta_i), i = 1, 2, \ldots, 9 \tag{7}$$

where $V_i$ is the reading of the $i^{\text{th}}$ interferometer, $V_i(\cdot)$ is the function of the length variation of laser path from the reference position to the measurement position, $(x_{di}, y_{di}, z_{di})$ is the installation position of the $i^{\text{th}}$ interferometer, and $(\alpha_i, \beta_i)$ is the angle of the $i^{\text{th}}$ laser beam.

Also, to make the model solvable, the number of equations is supposed to be larger than the number of unknowns. Hence, all the interferometers' readings are recorded at every 20 distinct measurement positions for one displacement computation. Let $(x_n, y_n, z_n, \theta_n, \varphi_n, \omega_n)$ be the displacement at the $n^{\text{th}}$ measurement positions, $n = 1, 2, \ldots, 20$, the nine interferometers' reading at 20 measurement positions are

$$V_{ni} = V_{ni}(x_n, y_n, z_n, \theta_n, \varphi_n, \omega_n, x_{di}, y_{di}, z_{di}, \alpha_i, \beta_i), \ n = 1, 2, \ldots, 20, \ i = 1, 2, \ldots, 9 \tag{8}$$

where $V_{ni}$ is the reading of the $i^{\text{th}}$ interferometer at the $n^{\text{th}}$ measurement position, $n = 1, 2, \ldots, 20$, $i = 1, 2, \ldots, 9$.

Since there are a total of 180 equations and 165 unknowns in Equation (8), the 6-DOF displacement measurement model (8) is also theoretically solvable. Similar to the analysis in Section 2.2, the installation errors do not affect the displacement computation, and the nonlinear model without any approximation provides high accuracy.

As we can see, both the 3-DOF model and the 6-DOF model are nonlinear equation systems with a large number of unknowns. In this case, the existence and uniqueness of the solution, as well as the solution algorithm, are all key factors affecting the accurate displacement computation, which are discussed in Section 3.

## 3. Multi-DOF Displacement Computation

The solution and the algorithm of the multi-axis interferometer measurement model are discussed. For simplicity, only the solution of the 3-DOF displacement measurement model (4) is discussed, and the solution principle of the 6-DOF model (8) is the same.

### 3.1. Problem of Solving Nonlinear Equation System

Rewriting the nonlinear equation system (4) as

$$\mathbf{F}(\mathbf{X}) = 0 \tag{9}$$

where $\mathbf{X}$ is the variable consisting of the 48 unknowns; $\mathbf{F}(\cdot)$ is the equation function of Equation (4).

The classical algorithm of the nonlinear equation system (9) is the Gauss–Newton method [16], in which the iteration formula is

$$\mathbf{X}^{(k+1)} = \mathbf{X}^{(k)} - \left[\mathbf{J}^T\left(\mathbf{X}^{(k)}\right)\mathbf{J}\left(\mathbf{X}^{(k)}\right)\right]^{-1}\mathbf{J}^T\left(\mathbf{X}^{(k)}\right)\mathbf{F}\left(\mathbf{X}^{(k)}\right) \tag{10}$$

where $k$ is the number of iterations and $\mathbf{J}(\cdot)$ is the Jacobi matrix of $\mathbf{F}(\cdot)$.

The problem is that in the computation process of solving (4) in the form of Equation (9) by using Equation (10), $\mathbf{J}\left(\mathbf{X}^{(k)}\right)$ is always going to be singular and $\mathbf{J}^T\left(\mathbf{X}^{(k)}\right)\mathbf{J}\left(\mathbf{X}^{(k)}\right)$ is not invertible due to its huge condition number, which makes it impossible to continue the iteration. In fact, for a nonlinear equation system with a singular Jacobi matrix, it is proven that the solution is not unique, and both the Newton method and Gauss–Newton method lose their convergence [16]. Hence, the improved algorithm needs to be applied. As it is proven that the solution of a nonlinear equation system with a singular Jacobi matrix is a set with at least one dimension and the Levenberg–Marquardt method (LM method) can be used to converge to a solution on the solution set [20], the iteration formula is

$$\mathbf{X}^{(k+1)} = \mathbf{X}^{(k)} - \left[\mathbf{J}^T\left(\mathbf{X}^{(k)}\right)\mathbf{J}\left(\mathbf{X}^{(k)}\right) + \mu\mathbf{I}\right]^{-1}\mathbf{J}^T\left(\mathbf{X}^{(k)}\right)\mathbf{F}\left(\mathbf{X}^{(k)}\right) \tag{11}$$

where $\mathbf{I}$ is the identity matrix and $\mu$ is the steepest descent factor.

However, although there are improved algorithms such as the LM method and Newton method based on the MP inverse mentioned in [21] for solving nonlinear equations with a singular Jacobi matrix, and since the solution of Equation (4) is not unique and the iteration varies with the initial iterate, there is still a deviation between the computation result of the 48 unknowns and their true value. As an illustration, the computation of solving Equation (4) with the Newton method is simulated: a set of true values of unknowns is given, and 10 different initial values are randomly generated. In contrast, the iterations of $x_1$ and $x_{d1}$ using the 10 initial values are shown in Figure 5. The detail of Figure 5 is drawn as Figure 6 to observe the convergence results.

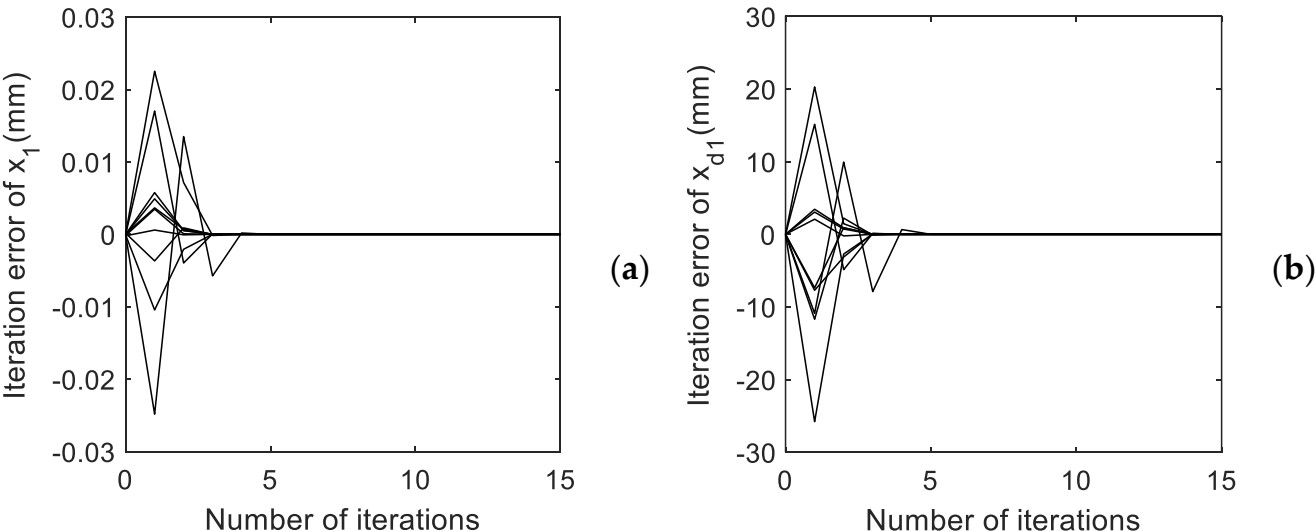

**Figure 5.** (**a**) Iteration error of $x_1$. (**b**) Iteration error of $x_{d1}$.

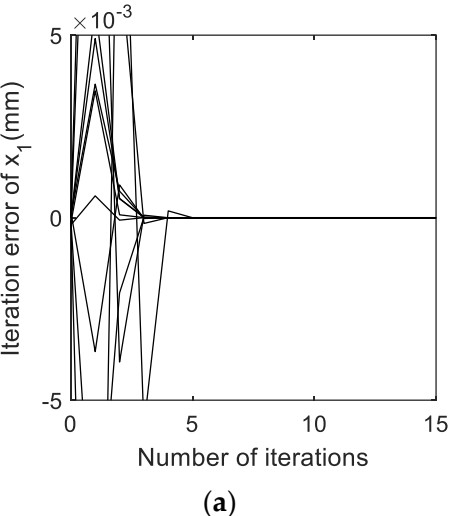 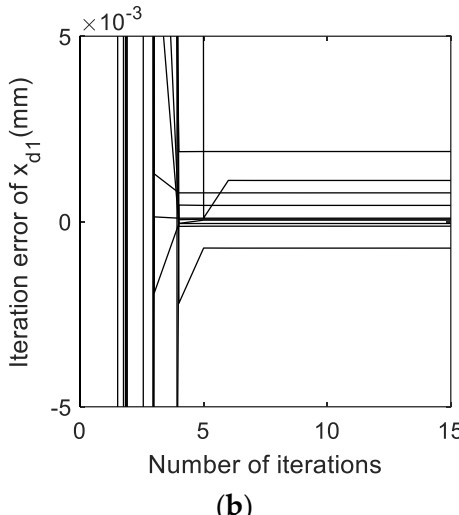

(a)  (b)

**Figure 6.** (**a**) Detail of iteration error of $x_1$. (**b**) Detail of iteration error of $x_{d1}$.

From Figure 5, the iterations of $x_1$ and $x_{d1}$ both converge; however, from Figure 6, the convergence of $x_{d1}$ changes with initial values. Hence, the convergence accuracy of $x_1$ and $x_{d1}$ are different, which means that the accuracy of each component of the equation system solution is different. Since we only care about the solution of displacement $(x_n, y_n, \theta_n), n = 1, 2, \ldots, 10$, can it be proven that all the displacement unknowns can be accurately solved? The answer is yes, and the proof is given in the following sections.

### 3.2. Solution Component Uniqueness Theory

As shown in Figure 6, the convergence of each variable of the nonlinear equation system with a singular Jacobi matrix is different, which is proven to be related to the uniqueness of the solution component in this section. To prove the solvability of displacement in the interferometer measurement model, the solution component uniqueness theory is proposed.

Without loss of generality, consider the nonlinear equation system

$$\mathbf{F}(\mathbf{X}) = \mathbf{0} \tag{12}$$

where $\mathbf{X} \in \mathbb{R}^N$, $\mathbf{F} : \mathbb{R}^N \to \mathbb{R}^M$ is a continuously differentiable nonlinear function.

Equation (12) is said to be a singular nonlinear equation system (SNES) if it has a singular Jacobi matrix. Let $\mathbf{X} = [x_1, \ldots, x_N]^T$ be any one in the solution set of Equation (12), the solution component $x_n$ is said to be unique or with uniqueness if there is only one value for it, where $n \in \mathbb{N}, 1 \le n \le N$.

Let $\mathbf{X}^* = [x_1^*, \ldots, x_N^*]^T$ be the true value of the unknowns in Equation (12). If we look at the interferometer model (4), $\mathbf{X}^*$ is the true value of the 48 unknowns. Let $\hat{\mathbf{X}} = [\hat{x}_1, \ldots, \hat{x}_N]^T$ be the solution computed by any improved algorithm. For the case that Equation (12) is SNES, as mentioned in Section 2, $\hat{\mathbf{X}}$ varies with the initial iterate, and the solution of Equation (12) is a set with at least one dimension. $\mathbf{X}^*$ and $\hat{\mathbf{X}}$ both belong to the solution set, but there is no evidence that $\mathbf{X}^* = \hat{\mathbf{X}}$ holds. The following Theorem 1 shows that the distance between $x_n^*$ and $\hat{x}_n$ depends on the solution component uniqueness.

**Theorem 1.** *For $n \in \mathbb{N}, 1 \le n \le N$, $\hat{x}_n = x_n^*$ holds if $\hat{x}_n$ is unique. Otherwise, $\hat{x}_n = x_n^*$ might not hold.*

**Proof.** For the condition that $\hat{x}_n$ is unique, there is only one value for the solution of $x_n$. Since both $\mathbf{X}^*$ and $\hat{\mathbf{X}}$ are the solution of (12), it is easy to check that $\hat{x}_n = x_n^*$ holds. In contrast, for the condition that $\hat{x}_n$ is not unique, there are two distinct solutions of (12) set as $\mathbf{X}_A$ and $\mathbf{X}_B$ with two distinct $n^{th}$ components set as $x_{nA}$ and $x_{nB}$. Without loss of generality,

suppose that $\mathbf{X}^* = \mathbf{X}_A$ and $\hat{\mathbf{X}} = \mathbf{X}_B$, then we have $x_n^* = x_{nA}$ and $\hat{x}_n = x_{nB}$, which indicate that $\hat{x}_n \neq x_n^*$ holds. Thus, the proof is finished.

Theorem 1 implies that only for the solution component with uniqueness, the result computed by the algorithm always converges to the true value. Therefore, for the aim to solve the 30 displacement unknowns in Equation (4), the uniqueness of the corresponding 30 solution components must be verified. The following Theorem 2 provides the basis to help verify the solution component uniqueness.

**Lemma 1.** *All the components of solution* $\mathbf{X}$ *are unique if* $\mathbf{J}\left(\mathbf{X}^*\right)$ *has full column rank.*

**Proof.** For the $\mathbf{J}\left(\mathbf{X}^*\right)$ with full column rank, it follows from the continuity of $\mathbf{F}(\mathbf{X})$ that there exists the neighborhood of $\mathbf{X}^*$ as $B\left(\mathbf{X}^*, r\right)$ with $r > 0$ such that $\text{rank}[\mathbf{J}(\mathbf{X})] = \text{rank}\left[\mathbf{J}\left(\mathbf{X}^*\right)\right]$ holds for all the $\mathbf{X} \in B\left(\mathbf{X}^*, r\right)$., i.e., for all the $\mathbf{X} \in B\left(\mathbf{X}^*, r\right)$, $\mathbf{J}(\mathbf{X})$ has full column rank. Thus, it follows from the Newton method [22] that $\mathbf{X}^{(k)}$ converges to the unique solution $\mathbf{X}^*$. Therefore, all the components of $\mathbf{X}$ converge to the unique solution, the proof is finished.

**Theorem 2.** *Let* $\mathbf{J}\left(\mathbf{X}^*\right) = [\mathbf{p}_1, \ldots, \mathbf{p}_N]$, *where* $\mathbf{p}_n \in \mathbb{R}^{M \times 1}$ *for* $n = 1, 2, \ldots, N$. *Then,* $x_n$ *is not unique if* $\text{rank}\left[\mathbf{p}_1, \ldots, \mathbf{p}_N\right] = \text{rank}\left[\mathbf{p}_1, \ldots, \mathbf{p}_{n-1}, \mathbf{p}_{n+1}, \ldots, \mathbf{p}_N\right]$ *holds. Otherwise,* $x_n$ *is unique.*

**Proof.** According to the definition of the Jacobi matrix, the total differential of (12) at $\mathbf{X}^*$ is

$$\sum_{j=1}^{N} \mathbf{p}_j \Delta x_j = \mathbf{0} \tag{13}$$

If $\text{rank}[\mathbf{p}_1, \ldots, \mathbf{p}_N] = \text{rank}\left[\mathbf{p}_1, \ldots, \mathbf{p}_{n-1}, \mathbf{p}_{n+1}, \ldots, \mathbf{p}_N\right]$ holds, there is

$$\mathbf{p}_n = \sum_{j=1, j \neq n}^{N} \alpha_j \mathbf{p}_j \tag{14}$$

where $\alpha_j \in \mathbb{R}$.

Substituting Equation (14) into Equation (13), then we have

$$\sum_{j=1, j \neq n}^{N} \mathbf{p}_j \left(\Delta x_j + \alpha_j \Delta x_n\right) = \mathbf{0} \tag{15}$$

Let

$$y_l = \begin{cases} x_l + \alpha_l x_n, 1 \leq l < n \\ x_{l+1} + \alpha_{l+1} x_n, n \leq l \leq N - 1 \end{cases} \tag{16}$$

Substituting Equation (16) into Equation (15), there is

$$\left[\mathbf{p}_1, \ldots, \mathbf{p}_{n-1}, \mathbf{p}_{n+1}, \ldots, \mathbf{p}_N\right] \cdot [\Delta y_1, \ldots, \Delta y_{N-1}]^T = \mathbf{0} \tag{17}$$

Taking (17) as the total differential of the equation system, $\mathbf{F}(\mathbf{Y}) = \mathbf{0}$ where $\mathbf{Y} = [y_1, \ldots, y_{N-1}]^T$, for which the Jacobi matrix is $[\mathbf{p}_1, \ldots, \mathbf{p}_{n-1}, \mathbf{p}_{n+1}, \ldots, \mathbf{p}_N]$.

Without loss of generality, suppose that $[\mathbf{p}_1, \ldots, \mathbf{p}_{n-1}, \mathbf{p}_{n+1}, \ldots, \mathbf{p}_N]$ has full rank, otherwise we can repeat the above process to cut down the column elements. Then, it follows

from Lemma 1 that all the solutions of $y_1$ are unique. Let $y_1^*$ be the unique solution of $y_1$, it follows from (16) that

$$
\begin{pmatrix}
1 & & & \alpha_1 & \\
& \ddots & & \vdots & \\
& & 1 & \alpha_{n-1} & \\
& & & \alpha_{n+1} & 1 \\
& & & \vdots & & \ddots \\
& & & \alpha_N & & & 1
\end{pmatrix}_{(N-1)\times N}
\begin{pmatrix}
x_1 \\
x_2 \\
\vdots \\
x_N
\end{pmatrix}
=
\begin{pmatrix}
y_1^* \\
y_2^* \\
\vdots \\
y_{N-1}^*
\end{pmatrix}
\tag{18}
$$

holds, which is the linear equation system of $[x_1, \ldots, x_N]^T$. Notice that the blanks in the matrix in (18) are all 0.

It is easy to check that $x_n$ in (18) has infinitely many solutions, which means $x_n$ is not unique. The proof of the first half of the statement is finished.

As for the other condition that $\mathrm{rank}[\mathbf{p}_1, \ldots, \mathbf{p}_N] > \mathrm{rank}\left[\mathbf{p}_1, \ldots, \mathbf{p}_{n-1}, \mathbf{p}_{n+1}, \ldots, \mathbf{p}_N\right]$, the discussion is divided into the following two cases: For the case that $\left[\mathbf{p}_1, \ldots, \mathbf{p}_{n-1}, \mathbf{p}_{n+1}, \ldots, \mathbf{p}_N\right]$ has full column rank, $[\mathbf{p}_1, \ldots, \mathbf{p}_N]$ must have full column rank since its rank is larger than $\left[\mathbf{p}_1, \ldots, \mathbf{p}_{n-1}, \mathbf{p}_{n+1}, \ldots, \mathbf{p}_N\right]$. Then, it follows from Lemma 1 that $x_n$ is unique. For the other case that $\left[\mathbf{p}_1, \ldots, \mathbf{p}_{n-1}, \mathbf{p}_{n+1}, \ldots, \mathbf{p}_N\right]$ has no full column rank, let $\left[\mathbf{p}_1, \ldots, \mathbf{p}_{n-1}, \mathbf{p}_{n+1}, \ldots, \mathbf{p}_N\right]$ $= [\mathbf{q}_1, \ldots, \mathbf{q}_W]$ after cutting down all the linearly dependent columns by taking the steps same as (14)–(17), where $[\mathbf{q}_1, \ldots, \mathbf{q}_W]$ has full column rank and $W < N - 1$. Then, the new Jacobi matrix becomes $\left[\mathbf{p}_n, \mathbf{q}_1, \ldots, \mathbf{q}_W\right]$, which has full column rank. Hence, according to Lemma 1, $x_n$ is unique. The proof of the second half of the statement is finished.

From Theorem 2, the uniqueness of the solution component can be determined by computing whether the corresponding column in the Jacobi matrix is independent. Therefore, to verify the uniqueness of the displacement solution in the interferometer model, the columns dependence computation methods are discussed in the following.

Although there are Gaussian elimination, determinant computation, and other numerical algorithms used to compute the dependence of vectors [23], since the numerical error exists in the Jacobi matrix computation, the dependence computation result might go wrong if the tolerance of the algorithms is improper (tolerance can only be selected according to empirical methods, without means of debugging or correction due to the lack of appropriate feedback). To make the dependence computation more accurate, a numerical algorithm for dependence computation based on principle component analysis (PCA) is proposed.

For the given Jacobi matrix $\mathbf{J}_{M \times N} = [\mathbf{p}_1, \ldots, \mathbf{p}_N]$, normalizing by

$$
\mathbf{A} = [\mathbf{p}_1 - \mathrm{mean}(\mathbf{p}_1), \ldots, \mathbf{p}_N - \mathrm{mean}(\mathbf{p}_N)]
\tag{19}
$$

The covariance matrix of $\mathbf{A}$ is

$$
\mathbf{C} = \mathbf{A}^T \mathbf{A} / M
\tag{20}
$$

Computing the eigenvalue decomposition

$$
\mathbf{C} = \mathbf{V} \mathbf{D} \mathbf{V}^T
\tag{21}
$$

where $\mathbf{V} = [\mathbf{v}_1, \ldots, \mathbf{v}_N]$ is the eigenvector matrix with $\mathbf{v} \in \mathbb{R}_{[-1,1]}^{N \times 1}$, $\mathbf{D} = \mathbf{diag}(\sigma_1, \ldots, \sigma_N)$ is a diagonal matrix whose diagonal elements are eigenvalues in descending order.

According to PCA, after setting the threshold $\mathrm{TH}_\sigma$ and $r$ with that $\sigma_1 > \cdots > \sigma_r > \mathrm{TH}_\sigma > \sigma_{r+1} > \cdots > \sigma_N$, we have

$$
\mathbf{J}_{\mathrm{prin}} = \mathbf{J} \cdot \mathbf{V}_r
\tag{22}
$$

for dimensionality reduction of $\mathbf{J}$, where $\mathbf{V}_r = [\mathbf{v}_1, \ldots, \mathbf{v}_r]$.

If $\mathbf{p}_n$ in $\mathbf{J}$ is strongly dependent, which indicates that it carries less information, we believe that it contributes less to $\mathbf{J}_{\text{prin}}$ so that the weight of $\mathbf{p}_n$ in Equation (22) is smaller. Suppose that $\mathbf{V}_r = [\mathbf{w}_1, \ldots, \mathbf{w}_N]^T$, $\mathbf{w} \in \mathbb{R}^{r \times 1}$, then the weight of $\mathbf{p}_n$ is $\mathbf{w}_n$. Thus, taking the infinite norm of $\mathbf{w}_n$ as the independent factor (IDF), that is

$$\text{IDF}_n = \|\mathbf{w}_n\|_\infty \tag{23}$$

There is $\text{IDF}_n \in [0,1]$ since $\mathbf{v} \in \mathbb{R}^{N \times 1}_{[-1,1]}$. The closer $\text{IDF}_n$ is to 0, which indicates that $\mathbf{p}_n$ has weaker independence and stronger dependence in $\mathbf{J}$, it follows from Theorem 2 that $x_n$ is more likely to be not unique; on the contrary, with stronger independence and weaker dependence of $\mathbf{p}_n$, $x_n$ is more likely to be unique.

According to the above, the IDF shows the possibility of whether a solution component is unique. After computing the IDFs of all the vectors in the Jacobi matrix, some simple classification algorithms such as the k-nearest neighbor algorithm [24] can be used to classify all the components into two categories according to the size of their IDFs, so as to determine the components' uniqueness. A simple approach is setting a threshold $\text{TH}_{\text{IDF}} \in (0,1)$, there is

$$x_n \begin{cases} \text{is not unique}, \text{IDF}_n < \text{TH}_{\text{IDF}} \\ \text{is unique}, \text{IDF}_n > \text{TH}_{\text{IDF}} \end{cases} \tag{24}$$

The brief process of the solution component uniqueness algorithm proposed above is summarized as follows.

So far, by using Theorem 1, Theorem 2, and Algorithm 1, it can be determined whether each unknown in the SNES can be solved accurately. Also, the solvability of the displacement in the interferometer model can be verified, which is discussed in the following Section 3.3.

---

**Algorithm 1:** Solution component uniqueness algorithm.

---

1 : **Initialization:** Given $\text{TH}_\sigma > 0$, $\text{TH}_{\text{IDF}} \in (0,1)$, $\mathbf{X}$ is a solution in the solution set of (12)
2 : $[\mathbf{p}_1, \ldots, \mathbf{p}_N] = \mathbf{J}(\mathbf{X})$
3 : $\mathbf{A} = [\mathbf{p}_1 - \text{mean}(\mathbf{p}_1), \ldots, \mathbf{p}_N - \text{mean}(\mathbf{p}_N)]$
4 : $\mathbf{C} = \mathbf{A}^T \mathbf{A}/M$
5 : Eigen decomposition $\mathbf{C} = \mathbf{V} \mathbf{D} \mathbf{V}^T$, where $\mathbf{V} = [\mathbf{v}_1, \ldots, \mathbf{v}_N]$, $\mathbf{D} = \text{diag}(\sigma_1, \ldots, \sigma_N)$
6 : Find $r$ satisfies $\sigma_1 > \cdots > \sigma_r > \text{TH}_\sigma > \sigma_{r+1} > \cdots > \sigma_N$
7 : $[\mathbf{w}_1, \ldots, \mathbf{w}_N]^T = [\mathbf{v}_1, \ldots, \mathbf{v}_N]$
8 : $[\text{IDF}_1, \ldots, \text{IDF}_N] = [\|\mathbf{w}_1\|_\infty, \ldots, \|\mathbf{w}_N\|_\infty]$
9: Compute the uniqueness of each component by Equation (24)

---

### 3.3. Principle of Displacement Computation

In this Subsection, the algorithm of the displacement computation of the interferometer measurement model is proposed, and the high accuracy of the displacement computation result is verified.

Since the classic Gauss–Newton method is not applicable to SNES, the Newton method with MP inverse is proposed to solve the displacement from the interferometer model (4). The linear expansion approximation of the equation system (12) at the given iteration $\mathbf{X}^{(k)}$ is

$$\mathbf{F}(\mathbf{X}^{(k)}) + \mathbf{J}(\mathbf{X}^{(k)})(\mathbf{X} - \mathbf{X}^{(k)}) = \mathbf{0} \tag{25}$$

In the Newton method, the solution of $\mathbf{X}$ in (25) is taken as the next step $\mathbf{X}^{(k+1)}$ in the iteration. In the case that $\mathbf{J}(\mathbf{X}^{(k)})$ is not a square matrix, the Gauss–Newton method solves $\mathbf{X}^{(k+1)}$ by multiplying Equation (25) by $[\mathbf{J}^T(\mathbf{X}^{(k)})\mathbf{J}(\mathbf{X}^{(k)})]^{-1}\mathbf{J}^T(\mathbf{X}^{(k)})$, which fails

when $\mathbf{J}^T\left(\mathbf{X}^{(k)}\right)\mathbf{J}\left(\mathbf{X}^{(k)}\right)$ is singular. In this paper, let Equation (25) be the inconsistent linear equation system of $\left(\mathbf{X} - \mathbf{X}^{(k)}\right)$, then the minimum 2-norm solution is

$$\mathbf{X} - \mathbf{X}^{(k)} = \mathbf{J}^\dagger(\mathbf{X}^{(k)})[-\mathbf{F}(\mathbf{X}^{(k)})] \tag{26}$$

where $\mathbf{J}^\dagger\left(\mathbf{X}^{(k)}\right)$ is the MP inverse of $\mathbf{J}\left(\mathbf{X}^{(k)}\right)$, which is calculated by singular value decomposition

$$\mathbf{J}^\dagger(\mathbf{X}^{(k)}) = \mathbf{V}_1\mathbf{S}_1^{-1}\mathbf{U}_1^T \tag{27}$$

where $\mathbf{V}_1, \mathbf{S}_1, \mathbf{U}_1$ satisfy

$$\mathbf{J}(\mathbf{X}^{(k)}) = \mathbf{U}\mathbf{S}\mathbf{V}^T = \begin{bmatrix}\mathbf{U}_1\\\mathbf{U}_2\end{bmatrix}^T\begin{bmatrix}\mathbf{S}_1 & \mathbf{0}\\\mathbf{0} & \mathbf{0}\end{bmatrix}\begin{bmatrix}\mathbf{V}_1\\\mathbf{V}_2\end{bmatrix} \tag{28}$$

where $\mathbf{S}$ is the diagonal matrix, whose diagonal elements are singular values in descending order.

Therefore, according to Equation (26), the iteration formula is

$$\mathbf{X}^{(k+1)} = \mathbf{X}^{(k)} + \mathbf{J}^\dagger(\mathbf{X}^{(k)})[-\mathbf{F}(\mathbf{X}^{(k)})] \tag{29}$$

The brief process of the proposed Newton method with MP inverse is summarized as follows.

Since the convergence of the Newton method with MP inverse is discussed [22], Algorithm 2 can be used to solve the interferometer model (4). However, the solution component uniqueness needs to be verified.

---

**Algorithm 2:** Newton method with MP inverse.

---

1 : **Initialization:** Given $k_{\max}$, $\mathbf{X}^{(0)}$, $k:= 0$
2 : **While** $k < k_{\max}$ **do**
3 : Singular Value Decomposition $\mathbf{J}\left(\mathbf{X}^{(k)}\right) = \mathbf{U}\mathbf{S}\mathbf{V}^T$
4 : Find $\mathbf{V}_1, \mathbf{S}_1, \mathbf{U}_1$ satisfy (28)
5 : $\mathbf{J}^\dagger\left(\mathbf{X}^{(k)}\right) = \mathbf{V}_1\mathbf{S}_1^{-1}\mathbf{U}_1^T$
6 : $\mathbf{X}^{(k+1)} = \mathbf{X}^{(k)} + \mathbf{J}^\dagger\left(\mathbf{X}^{(k)}\right)\left[-\mathbf{F}\left(\mathbf{X}^{(k)}\right)\right]$
7:    $k = k + 1$
8: **end while**

---

Algorithm 1 is used to compute the solution uniqueness of the 48 unknowns in Equation (4). Number the 48 unknowns as

$$\begin{aligned}\mathbf{X} = \quad & [x_{d1}, x_{d2}, \ldots, x_{d6}, x_1, x_2, \ldots, x_{10},\\ & y_{d1}, y_{d2}, \ldots, y_{d6}, y_1, y_2, \ldots, y_{10},\\ & \alpha_1, \alpha_2, \ldots, \alpha_6, \theta_1, \theta_2, \ldots, \theta_{10}]\end{aligned} \tag{30}$$

Then, the eigenvalues $[\sigma_1, \ldots, \sigma_{48}]$ in Equation (21) are shown in Figure 7a, and the IDFs of the 48 vectors in the Jacobi matrix are shown in Figure 7b. It is easy to check from Figure 7b that the IDF of the vectors whose numbers are 1, 2, 4, 5, 7, 8, 10, 11, 13, 14, 16, and 17 are much lower than the others. The corresponding solution components are not unique for the threshold, which is taken as $\mathrm{TH}_{\mathrm{IDF}} = 0.3$. Thus, the convergence results of these unknowns are not accurate, which means that the computation of the interferometers' installation position $(x_d, y_d)$ is not accurate. On the contrary, the other 36 unknowns, including the 3-DOF displacement, are solved with high accuracy. The solvability of the displacement in the interferometer model is proven.

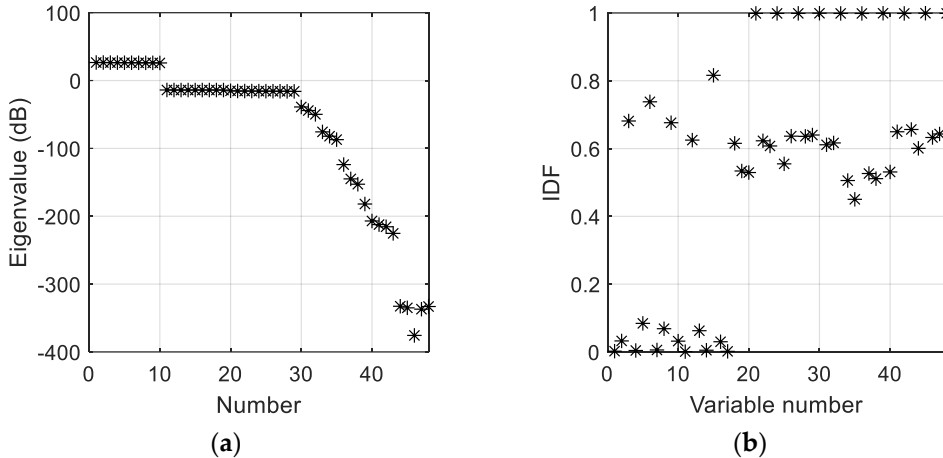

**Figure 7.** (**a**) Eigenvalues of covariance matrix. (**b**) Independent factors. The symbol * in the figure represents the data point.

## 4. Simulation Results

The 3-DOF displacement measurement of the proposed interferometer is simulated to verify the high accuracy. To verify the advantage that there is no measurement error caused by installation errors, the deviation between the nominal value and the true value of the interferometers' installation positions and orientations is set as shown in Table 1.

**Table 1.** Installation errors of lasers.

| Laser Number | 1 | 2 | 3 | 4 | 5 | 6 |
|---|---|---|---|---|---|---|
| $x_d(\times 10^{-3}$ mm) | −0.96 | 0.61 | −1.40 | 0.67 | 1.10 | −0.74 |
| $y_d(\times 10^{-3}$ mm) | −0.21 | 0.23 | −1.10 | 2.60 | −0.48 | 0.38 |
| $\alpha(\times 10^{-3}$ mrad) | 0.51 | −1.20 | 0.09 | −1.70 | −1.00 | 0.91 |

The simulation scheme is shown in Figure 8. A total of 10 groups of 3-DOF displacement in the range of the interferometer are randomly selected as the true values. By substituting the true values of displacements and the true values of installation parameters into the interferometer model (4), the readings of the six interferometers are simulated. Then, substituting the readings into Equation (4) and taking the displacements and the installation parameters as unknowns, the displacement computation is simulated by using the proposed method in Section 3. In solving Equation (4) with Algorithm 2, the nominal value of the installation parameters is taken as the initial iteration, and the initial iteration of the displacements is the computation result in the last time step.

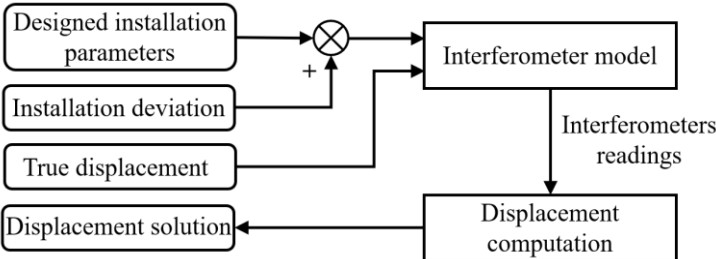

**Figure 8.** Scheme of the simulation process.

As the simulation results, the iteration error curves of the 48 unknowns are shown in Figures 9–12, where Figures 9 and 10 are the iteration error curves of the installation position $(x_d, y_d)$ and orientation $\alpha$ of the 6 interferometers; Figures 11 and 12 are the iteration error

curves of the 10 sets of displacement $(x, y, \theta)$. It is easy to check from Figures 9–12 that the 3-DOF displacement $(x, y, \theta)$ converges to the true value while the laser installation position $(x_d, y_d)$ does not. Hence, the solvability of the displacement is verified.

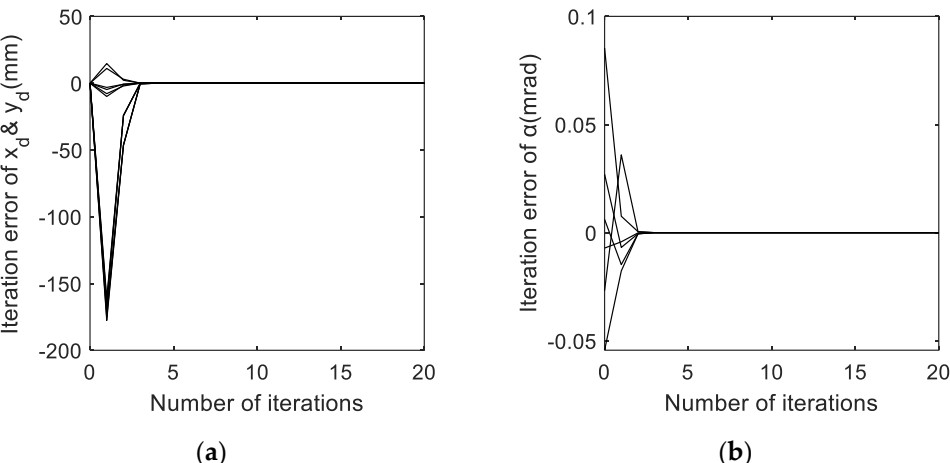

**Figure 9.** (**a**) Iteration error of $x_d$ and $y_d$. (**b**) Iteration error of $\alpha$.

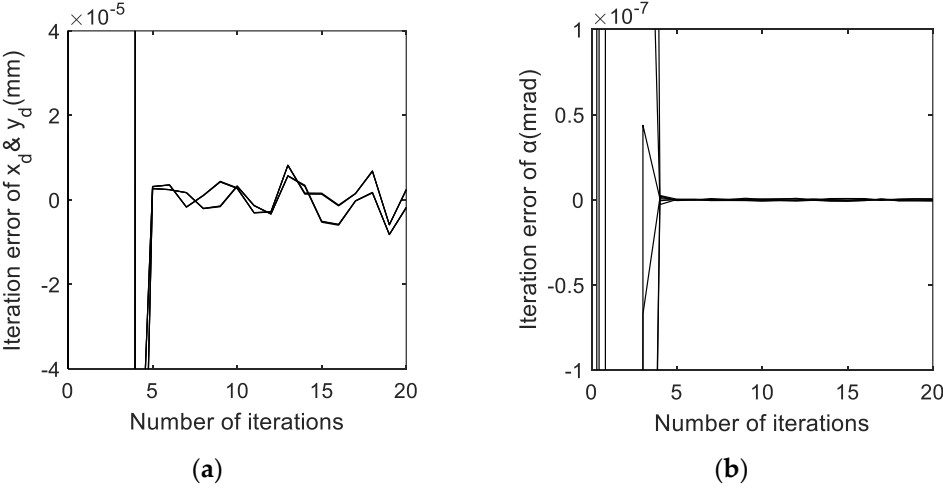

**Figure 10.** (**a**) Detail of iteration error of $x_d$ and $y_d$. (**b**) Detail of iteration error of $\alpha$.

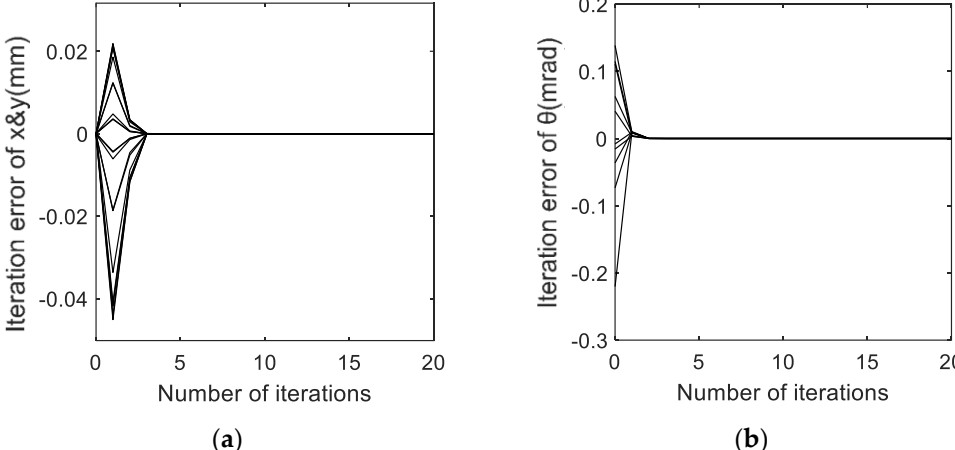

**Figure 11.** (**a**) Iteration error of $x$ and $y$. (**b**) Iteration error of $\theta$.

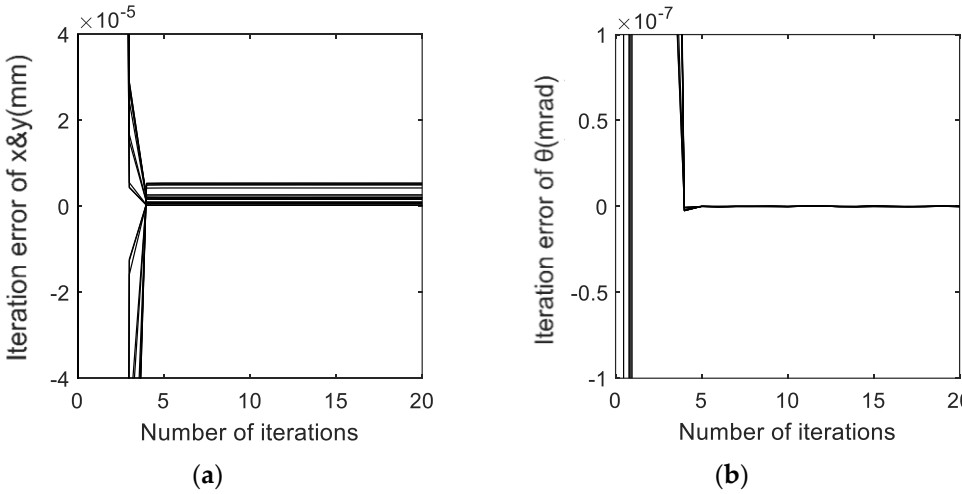

**Figure 12.** (**a**) Detail of iteration error of $x$ and $y$. (**b**) Detail of iteration error of $\theta$.

Further, the error reduction ratio (ERR) is defined as follows to verify the conclusion in Section 3.3 that only the unknowns with high IDF can be solved accurately: for the general nonlinear equation system $\mathbf{F}(\mathbf{X}) = \mathbf{0}$ where $\mathbf{X} = [x_1, \dots, x_N]^T$, the ratio of the variable error after iteration to the variable error before iteration is defined as ERR

$$\mathrm{ERR}_n = -20 \log \left| \frac{\hat{x}_n - x_n^*}{x_n^{(0)} - x_n^*} \right| \tag{31}$$

where $\mathrm{ERR}_n$ is the ERR of $x_n$, $x_n^*$ is the true value, $x_n^{(0)}$ is the initial iteration, and $\hat{x}_n$ is the convergence solution.

For 3-DOF displacement computation, namely the solution of Equation (4), there are a total of 48 unknowns including displacements and installation parameters. Arrange the 48 unknowns as Equation (30), the ERR of each unknown is shown in Figure 13. It can be seen from Figure 13 that the magnitude of ERRs corresponds to the IDFs shown in Figure 7b, which verifies that only the iteration of unknowns with a high IDF can converge to the true value.

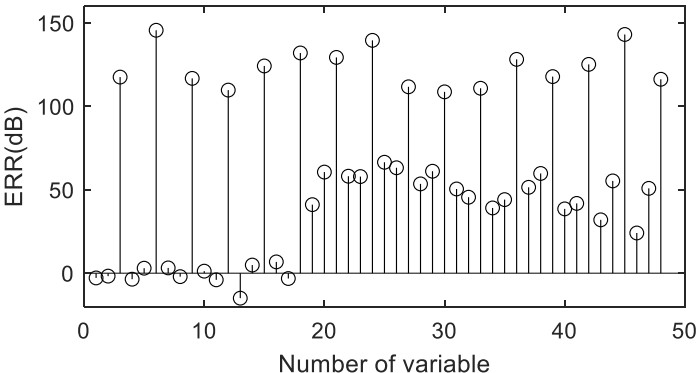

**Figure 13.** ERRs of the 48 unknowns.

To test the numerical accuracy of the proposed algorithm, 200 displacement measurements are simulated for the case that there is noise in the readings of interferometers, and the root mean square (RMS) of noise is set as 0.1 nm to simulate the practical engineering. The measurement error of displacement is shown in Figure 14, from which the mean and the standard deviation (STD) of displacement $x$ error are 0.155 nm and 1.884 nm, and the mean and STD of displacement $\theta$ error are $3.549 \times 10^{-8}$ mrad and $5.871 \times 10^{-7}$ mrad.

According to the above, the simulation results show the unbiasedness, low uncertainty, and high convergence speed of the displacement computation.

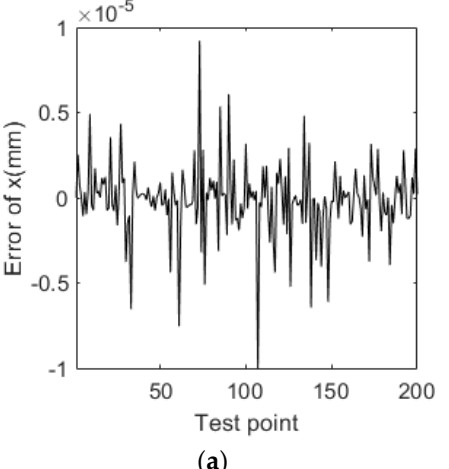
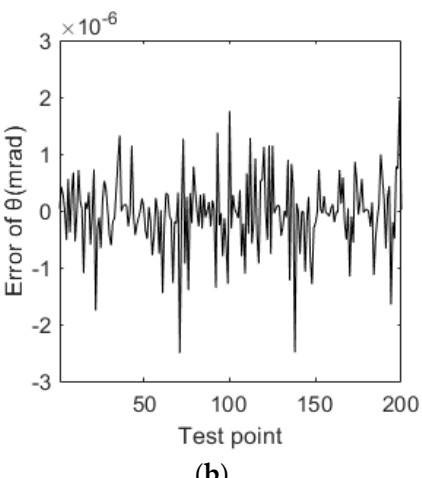

(**a**)
(**b**)

**Figure 14.** (**a**) Simulation result of *x* with reading noise (RMS = 0.1 nm). (**b**) Simulation result of *θ* with reading noise (RMS = 0.1 nm).

It should be noted that according to Algorithm 2 in Section 3.3, the amount of computation is proportional to the number of iterations, equations, and unknowns, which depend on the number of interferometers and measurement positions. Therefore, compared with traditional linear multi-axis interferometer models, the proposed iterative-based modeling and solving algorithm has a greater computational complexity. Nevertheless, in practical engineering, the computational time can be decreased by adopting a high-efficiency processing unit such as the FPGA with a parallel computing mode.

## 5. Conclusions

In this paper, a multi-axis laser interferometer is proposed, whose measurement is not affected by the installation errors. The interferometer measurement model shows that there is no need for the accurate installation of lasers, and the displacement computation shows the irrelevance between high measurement accuracy and installation errors, which is proven by the proposed solution component uniqueness theory for nonlinear equation systems. The simulation shows that even with installation errors close to 1 μm and 1 μrad, the interferometer is still able to measure the multi-DOF displacement accurately. For the case of where there is reading noise with RMS=0.1 nm, the STD of the displacement error is 1.884 nm and $5.871 \times 10^{-7}$ mrad.

This paper provides a novel way for the design and development of multi-axis interferometers, that is, with today's more and more developed computer processing ability, using a high-performance numerical computation to compensate for the lack of low mechanical manufacturing and assembly level, so that the measurement accuracy approaches the theoretical limit. The modeling method proposed is not limited to the topology of the interferometer in the paper, but can also be applied to other types of multi-axis interference systems. It is only necessary to take the installation parameters such as the laser angle as unknowns into the optical path model, and then the nonlinear computation method can be used to solve the model to obtain accurate displacement measurement results.

In the future, experiments will be designed to verify the measurement performance of the proposed interferometer. The optimized iteration algorithms with a lower complexity and higher speed will also be studied.

**Author Contributions:** Writing—original draft, T.W.; Writing—review and editing, J.H., Y.Z. and G.H. All authors have read and agreed to the published version of the manuscript.

**Funding:** This research was funded by the National Natural Science Foundation of China, grant number 51175296 and 51677104, and by the Autonomous Scientific Research Project of the State Key Laboratory of Tribology, Tsinghua University, grant number SKLT2018C07.

**Institutional Review Board Statement:** Not applicable.

**Informed Consent Statement:** Not applicable.

**Data Availability Statement:** Not applicable.

**Conflicts of Interest:** The authors declare no conflict of interest.

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
