# Peer review of "Multi-Axis Laser Interferometer Not Affected by Installation Errors Based on Nonlinear Computation"

_applsci, doi:10.3390/app131910887_

Round 1

Reviewer 1 Report

Dear, I can't read and understand what you wrote.  I would like to review your manuscript after it have been rewriten with a help of a native speaker.

English is very poor and the manuscript must be rewriten with the help of a native speaker. 

Author Response

Thanks for your review. The English writing is improved in the revision.

Reviewer 2 Report

 1-     The abstract must be contained the important gain results

2-      The novelty in the paper is not clear, and the difference in this work compared to others must be inserted in the introduction part.

3-     The conclusion and future expected work must be informed with more details 

Author Response

Thanks for your review. Here are the responses to your comments:

  1. The abstract is rewritten. The important gain result is that the proposed interferometer can reach high measurement accuracy even if there are installation errors.
  2. The second paragraph of the introduction introduces the existing works, the third paragraph introduces the problems existing in these works, and the fourth paragraph explains the difference between the proposed method and other works.
  3. The future expected work is added to the conclusion.

Reviewer 3 Report

The topic of the manuscript is relevant, the text is generally well written and well structured. The following points should be carefully supplemented to be published as a research article in Applied Sciences:

1) It should be discussed in the Introduction to what extent the approach to computer simulation and design of interferometers suggested by the authors is unique and original, possibly some references to similar researches should be included. 

2) The article presents only computer simulation results. The significance and relevance of the study for practical applications and possible ways of implementation should be discussed in the Conclusion in more detail.

Author Response

Thanks for your review. Here are the responses to your comments:

  1. To our knowledge, there is currently almost no research on introducing nonlinear computation into the solution of interferometer model. The existing modeling and solving of multi-axis interferometers are all based on structural optimization and linearization, and the typical works are given in the second paragraph of the introduction.
  2. The description about the significance and implementation is added to the conclusion. Please see the second paragraph of the conclusion: This paper provides a novel way for the design and development of multi-axis interferometers, that is, in today's more and more developed computer processing ability, using high-performance numerical computation to compensate for the lack of low mechanical manufacturing and assembly level, so that the measurement accuracy approaches the theoretical limit. The modeling method proposed is not limited to the topology of the interferometer in the paper, but can also be applied to other types of multi-axis interference systems. It is only necessary to take the installation parameters such as laser angle as unknowns into the optical path model, and then the nonlinear computation method can be used to solve the model to obtain accurate displacement measurement results.

Reviewer 4 Report

A multi-axis interferometer based on nonlinear computation was proposed, which avoided the measurement error caused by installation deviation by taking laser position and orientation as unknowns into measurement model and discussing the solution component uniqueness of nonlinear equation system. Several simulations were carried out to verify the reliability and accuracy of the proposed method. Some comments are given as follows.

1. It is better to verify the proposed method through experiments.

2. After reading the whole paper, it is still confused why use so many interferometers to measure displacement? Two interferometers are sufficient to measure the displacement of a two-dimensional moving table. If it is considered that the motion errors of the moving table will affect the measurement accuracy, a laser interferometer with multi-laser beams, such as XD laser from API and XM-60 from Renishaw, can be used to simultaneously measure the displacement and the motion errors. After that a compensation can be carried out to reduce the effects of the motion errors on the measurement accuracy of the displacement.

So why use six laser interferometers, as shown in Fig. 2, to measure the displacement of a two-dimensional moving table? How to control and measure the relative position and orientation errors between each interferometer? How does the relative position and orientation errors affect the measurement accuracy? Such a measurement method complicates the measurement and calculation of the displacement.

At the same time, with so many laser interferometers, how to consider the effects of each laser interferometer's stability, precision, and the errors caused by dead path on the measurement results?

Furthermore, this measurement method is costly.

3. Why the laser beam emits from the interferometer is not perpendicular to the measured stage, as shown in Fig. 1? In addition, was the laser emits from the interferometer directly reflected by the surface of the stage rather than a retroreflector? If so, what is the measurement principle of the interferometer? What are the requirements for the surface of the stage?

4. It seems that the results shown in Fig. 6, Fig. 10 and Fig. 12 are out of the scale of the vertical axis.

5. “To verify the advantage that there is no measurement error caused by installation deviation, the deviation between the designed value and the true value of the interferometers’ installation positions and orientations is set as shown in Table 1.” What are the designed value and the true value of the interferometers’ installation positions and orientations? How to measure the true value of the interferometers’ installation positions and orientations?

Author Response

Thanks for your review. Here are the responses to your comments:

  1. It is added to the conclusion section that the experiments will be designed in the future. Nevertheless, we believe that this paper provides a novel way for the design and development of multi-axis interferometers, that is, using high-performance numerical computation to compensate for the lack of low mechanical manufacturing and assembly level, which is of great value for publication.
  2. First, the multi-laser beams are necessary for compensating the motion errors you mentioned. However, no matter using multi-interferometers or an interferometer with multi-laser beams, once multi-laser beams are used, the relative position and orientation errors become a new problem that affect the measurement (The multi-interferometers and an interferometer with multi-laser beams are the same in principle. Even in XD laser from API and XM-60 from Renishaw, it is hard to control the relative position and orientation errors of each laser beam). Therefore, the goal of this paper is exactly abandoning the requirement of precision installation and assembly of multi lasers (You can see we take the relative position and orientation as unknowns instead of nominal values into the interferometer model), by using nonlinear computation method, the high accuracy measurement is still realized. We believe that in today's more and more developed computer processing ability, it is meaningful that using high-performance numerical computation to compensate for the lack of low mechanical manufacturing and assembly level.

The reason for why use six laser interferometers is added to the end of Section 2.2, the number of interferometers must guarantee that there is enough information to solve the equations. Besides, as mentioned above, the relative position and orientation errors of each interferometer will not affect the measurement accuracy, since we take them as unknowns instead of using nominal values to modeling.

  1. Ideally, the laser beam should be perpendicular to the stage, however the perpendicular is not perfect in practice due to the slight rotation of stage and the laser orientation error you mentioned in the second comment. In Fig. 1, the angle of deviation from the perpendicular is drawn to show this error, which will not affect the measurement of the proposed multi-axis interferometer.

The laser emits from the interferometer should reflects at the smooth surface of the stage or the retroreflector fixed on the stage. For simplicity, the retroreflector is not drawn in Fig. 1. To avoid the possible confusion, the related description is added to Section 2.1.

  1. Fig. 6, Fig. 10, and Fig. 12 are the zoom in picture of Fig. 5, Fig. 9, and Fig. 11. The goal of zoom in is to observe the convergence of iteration. For example, you can see x_1 and x_d1 both converge in Fig.5, however, after zooming in you can see that the convergence error of x_1 and x_d1 are different.
  2. The designed value of the interferometers’ installation positions and orientations is a nominal reference value, which can be obtained by measuring directly by using any device such as encoders. The true value is the real physics size, which cannot be obtain in practice since any measurement exist error. But in simulation the true value can be set by us. In the revision, all the “designed value” are replaced by “nominal value”.

Reviewer 5 Report

The manuscript presents a multi-DoF decoupling model and its nonlinear computation method. The authors claim that the model is unaffected by the interferometer installation deviation. And the nonlinear computation method is achieved without any linearization. The overall scheme is interesting. However, there are many issues should be explained and major revised before it can be accepted.

1. The title of the manuscript doesn’t express the content clearly. And the English expression should be polished by native speakers.

2. The whole method requires massive multi-DoF posture information firstly. And the multi-DoF can be computed iteratively. It seems the measurement can not be real-time achieved, however, which is a basic requirement for the semiconductor industry. Please explain the dynamic capability of the method.

3. In Fig. 9 to 12, the 3-DoF displacement converges while the interferometer installation error does not. What is the theoretical basis behind it?

4. The simulation result is based on a SNR of 60 dB. What is the exact meaning of this SNR here? If the displacement is in the order of nm level, does it mean that the error is only fm order? In the practical engineering, the accuracy of the interferometer can be limited to the order of 0.1 nm in the vacuum and the order of nm in the air. In these case, how about the multi-DoF decoupling accuracy of the proposed method?

5. The 3-DoF measurement model requires all the laser beams in the XOY plane. It is extremely difficult to achieve in practical situations. The model should be further extended to include these installation errors.

The title of the manuscript doesn’t express the content clearly. And the English expression should be polished by native speakers.

Author Response

Thanks for your review. Here are the responses to your comments:

  1. The title is modified as “Multi-axis Laser Interferometer Not Affected by Installation Errors Based on Nonlinear Computation”, which express that by using nonlinear computation, the proposed multi-axis interferometer can still measure accurately even if there are installation errors. English writing is improved in the revision.
  2. The description about the computational complexity is added to the end of Section 4. The proposed iterative based modeling and solving algorithm has a greater computational complexity. But the computational time can be decreased by adopting high efficiency processing unit such as the FPGA with parallel computing mode in practical engineering.
  3. It is because the constraints of the equation system are not sufficient to uniquely determine all the unknowns. Only the unknowns with unique solution can converge after the iteration. That is why this paper propose the solution component uniqueness theory to judge which unknowns are solvable and with unique solution. Please refer to Section 3.2 for relevant content.
  4. This is a good point. We are unable to know the real level of noise solely based on SNR. Therefore, in the revision, we use root mean square (RMS) of noise instead of SNR, and the simulation results are updated.
  5. We do have the more precision 6-DOF model discussed in Section 2.3, but the 3-DOF model is focused. On the one hand, simple 3-DOF model is more helpful to explain the modeling and solution methods proposed, which are easy to be extended to the 6-DOF model. On the other hand, because errors out of the horizontal plane are easier to control than errors in the horizontal plane (e.g., using horizontal calibration), the 3-DOF model is acceptable in some application where the requirement of accuracy is not too high.

Round 2

Reviewer 1 Report

1. I see that the authors have changed the title of paper. but both the former and the latter are still not suitable for the contents they are writing.

2. I agree that the paper being considered for the contents and the research results.

3. as I comment on the quality of the English, infact, the writing is very poor and the authors haven't have the manuscript polished since last review. so I strongly suggest that the authors doing so.

Infact, the writing is very poor. I strongly recommended that the manuscript being polished by a native speaker. only after that the paper can be considered for the publication.

Author Response

Thank you for your review. All the English grammar mistakes are corrected, and some expresses are improved.

Reviewer 4 Report

The author did not respond directly to all suggestions and comments.

In addition, I don't think in order to solve the equations to increase the number of interferometers is a good approach.

Minor editing of English language required.

Author Response

Thank you for your patient review. All the English grammar mistakes are corrected, and some expresses are improved.

In fact, the multi-axis measurement information can be obtained by building a mirror system with one laser source and multiple photoelectric detectors. Hence there is no need for multiple independent interferometers, which can greatly reduce structural complexity.

Reviewer 5 Report

The manuscript can be accepted now.

Author Response

Thank you for your review.